# UNBIASED TOKEN PRUNING FOR EFFICIENT LARGE MULTIMODAL MODELS

## ABSTRACT

Large Multimodal Models (LMMs) have demonstrated exceptional capabilities and drawn increasing attention. However, their substantial computational cost poses significant challenges for real-world applications. A considerable portion of this arises from the lengthy sequences of image tokens, bringing quadratically increasing computations due to the Transformer architecture. In light of this, recent works have explored visual token pruning for higher efficiency. Despite effective, they generally suffer from the inaccurate importance estimation (*i.e.*, what to prune) and the suboptimal pruning layers (*i.e.*, where to prune). This leads to notable visual information loss and inferior performance. In this work, we present an Unbiased Token Pruning (UTP) method to tackle these issues. For what to prune, we introduce an Unbiased Relevance Estimation (URE) strategy, which disentangles the interference of position embedding for more accurate importance assessment of visual tokens. For where to prune, we propose an Unbiased Token Retention (UTR) strategy, which solves the optimal pruning scheme by formulating the objective of minimizing the information loss as an integer linear programming problem. Extensive experiments demonstrate that our method outperforms existing state-of-the-art works and exhibits favorable performance in various tasks, showing its superiority for efficient inference of LMMs. Code will be publicly available.

## 1 INTRODUCTION

Recently, the field of computer vision has witnessed the significant advancements with the emergence of Large Multimodal Models (LMMs) (Liu et al., 2024c;a; Bai et al., 2025; Comanici et al., 2025; Zhu et al., 2025). These models integrate multi-modality input into the Large Language Models (LLMs) (Achiam et al., 2023; Dubey et al., 2024; Yang et al., 2025), thereby enabling them to process diverse types of information. Typically, LMMs embed the image and sentence into visual and textual tokens for LLMs. Leveraging the powerful generation and reasoning abilities of LLMs, they exhibit superior performance across various tasks (Li et al., 2024a; Liu et al., 2023).

However, the remarkable capabilities of LMMs are accompanied with high computation and memory costs (Jin et al., 2024), which impede their large-scale deployment in practical applications. Many studies have explored reducing the model size for enhanced efficiency by training LMMs with smaller LLMs (Shao et al., 2024; Hinck et al., 2024). However, these require considerable resource for training and are not applicable to the off-the-shelf LMMs. Recent works have dedicated to pruning visual tokens in the forward process during prefilling in a *training-free* manner, eliminating the notable redundancy in visual input for acceleration (Chen et al., 2024b; Shang et al., 2024; Chen et al., 2024a). In general, they consist of two steps: 1) estimating the relevance of visual tokens to the text prompt and ranking them based on the metric, *i.e.*, addressing *what to prune*, and 2) selecting the pruning layers and numbers to filter out the lowest-ranked tokens, *i.e.*, determining *where to prune*. For example, FastV Chen et al. (2024b) adopts the attention scores of the last token to preceding visual tokens as the importance metric and prunes less important tokens at a fixed layer in LLM.

While effective, existing approaches often face two generic drawbacks. First, regarding what to prune, we observe that the attention-based importance metric used in prior works (Chen et al., 2024b) tends to preserve tokens in certain positions, leading to an *attention bias* in estimating the relevance of visual tokens. For instance, as shown in Fig. 1.(a), visualization results from FastV reveal that the importance scores estimated based on LLM attention exhibit a strong correlation with token positions.

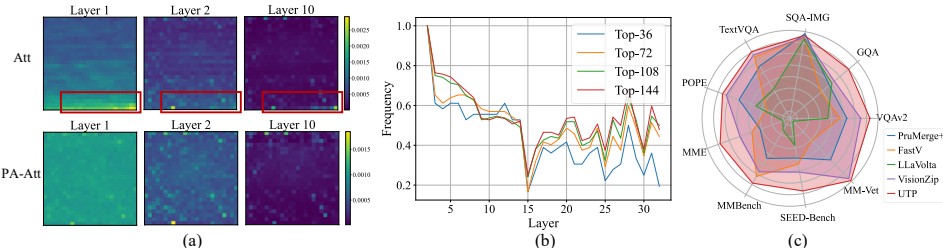

Figure 1: (a) The importance scores of visual tokens calculated by the attention-based metric (Att) in previous research, like FastV (Chen et al., 2024b) and our proposed metric (PA-Att), respectively. We employ LLaVA-1.5-7B (Liu et al., 2024a) model and 100 random image-text pairs sampled from various datasets for visualization. We observe that visual tokens positioned near the bottom right corner typically exhibit higher importance in Att and our PA-Att can well eliminate such bias. (b) The overlap frequency between important visual tokens in the to-be-pruned layer and those in other layers given original LMM. We take pruning at the second layer for example following Chen et al. (2024b). We examine various Top-$K$ conditions to provide a comprehensive analysis. We note the subsequent layers possess different important visual tokens from those of the to-be-pruned layer. (c) Comparison between UTP and others. UTP achieves state-of-the-art performance, showing superiority.

Specifically, visual tokens near the bottom right corner consistently receive higher attention scores compared with others. As a result, tokens in certain positions may be mistakenly preserved, even if they don't convey meaningful visual information. Secondly, regarding where to prune, existing works (Chen et al., 2024b;a; Shang et al., 2024) typically assume that important tokens are similarly distributed across layers, leading to pruning in a single fixed layer. Note that once a token is pruned at a certain layer, it will not exist in subsequent layers. However, as illustrated in Fig. 1.(b), later layers do not share the same important visual tokens as the selected layer for pruning. Consequently, pruning at a fixed layer introduces *layer bias*, leading to notable visual information loss in subsequent layers and ultimately suboptimal token retention.

In this work, we propose a training-free **U**nbiased **T**oken **P**runing (UTP) method to address these two critical issues. First, to tackle the attention bias for what to prune, we introduce a novel unbiased relevance estimation strategy based on the position-agnostic attention metric (PA-Att). Its core idea is to eliminate the influence of the position information during the token relevance estimation. To this end, we apply a uniform position embedding for all visual tokens, resulting in equal effects of position when deriving the importance metric by attention score. This thus provides more accurate importance assessment. Besides, for the mitigation of layer bias during determining where to prune, we propose an unbiased token retention scheme to reduce the redundancy of visual token in the forward process of prefilling while maintaining overall visual information as much as possible. It starts with measuring the pruning loss tolerance to assess the degree of visual information loss at each layer. The challenge of deriving token retention recipe is then reformulated as an integer linear programming problem with the layer-wise pruning token number as the independent variable to minimize the disturbance to the model. Using an integer programming solver (Huangfu & Hall, 2018), we can conveniently derive the layer-wise pruning token numbers offline without the extra inference overhead. We conduct extensive experiments and analyses across various benchmarks. As demonstrated in Fig. 1.(c), our method significantly outperforms existing works, achieving state-of-the-art performance. Besides, it accelerates inference speed notably, showing promising application potential.

## 2 RELATED WORK

**Large multimodal models.** The evolution of Large Multimodal Models (LMMs) signifies a further unlocking of the potential of Large Language Models (LLMs), enhancing their generation and reasoning capabilities for multimodal inputs (Bai et al., 2025; Zhu et al., 2025; Comanici et al., 2025). Leveraging the LLMs such as GPT-4 (Achiam et al., 2023), LLaMA (Touvron et al., 2023a;b; Dubey et al., 2024), and Qwen (Bai et al., 2023a; Qwen et al., 2025; Yang et al., 2025), which are pretrained on the extensive text corpora, LMMs like GPT4-V (Achiam et al., 2023), LLaVA (Liu et al., 2024c;a), and Qwen-VL (Bai et al., 2023b; 2025), *etc.*, have demonstrated exceptional performance across various visual-language tasks (Fu et al., 2023; Li et al., 2024a; Yu et al., 2023).

**Efficient inference of LMMs.** Despite achieving superior performance, existing LMMs generally suffer from the intensive computational cost and memory footprint, which greatly hinder their efficient deployment for practical scenarios. In light of this, many efforts have been directed to enhancing the inference efficiency of LMMs (Li et al., 2024b; Hu et al., 2024). For example, LLaVA-Phi (Zhu et al., 2024) and TinyLLaVA (Zhou et al., 2024) leverage the Phi-2 (Javaheripi et al., 2023) as the language component for a compact yet powerful architecture. MobileVLM (Chu et al., 2023) and MobileVLM-v2 (Chu et al., 2024) explore mobile-oriented architectural designs and training scheme. Vary-toy (Wei et al., 2024) and MoE-LLaVA (Lin et al., 2024) incorporates enhanced vision vocabulary and mixture of experts to enhance performance and efficiency, respectively.

**Visual token pruning.** Before the evolution of LMMs, there have been works (Rao et al., 2021; Bolya et al., 2022; Liang et al., 2022) to explore visual token pruning for Vision Transformers (ViTs), which show promising results and enhanced efficiency. For LMMs, a considerable portion of inference burden comes from the large number of embedded visual tokens in input. For example, LLaVA (Liu et al., 2024a) and Fuyu (Bavishi et al., 2023) introduce 576 and 1296 visual tokens for the image, respectively. Therefore, recent works have investigated visual token pruning for accelerating LMM's inference (Chen et al., 2024b; Shang et al., 2024; Chen et al., 2024a; Ye et al., 2024; Yang et al., 2024; Xing et al., 2024). For example, FastV (Chen et al., 2024b) leverages the attention score as the visual token importance metric and filters out those with less information. LLaVA-PruMerge (Shang et al., 2024) identifies spatial redundancy based on similarity for visual token reduction. Despite achieving promising results, they generally suffer from inaccurate importance estimation and suboptimal token retention, leading to notable information loss and inferior performance.

## 3 METHODOLOGY

In this section, we first delve into the basic implementation of LMMs in Sec. 3.1. To address the critical challenge of what visual tokens to prune, we introduce the unbiased relevance estimation in Sec. 3.2 to accurately assess their importance, aiming to retain the most crucial ones. Then, to decide where to prune visual tokens, we present unbiased token retention in Sec. 3.3 to select the optimal pruning scheme, thereby minimizing the impact of visual information loss on model performance. The outcome of these explorations is our unbiased token pruning (UTP) method for efficient inference of LMMs. Fig. 2 presents its overview.

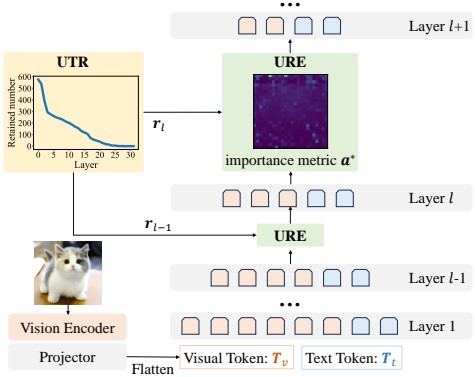

Figure 2: The overview of our UTP method.

### 3.1 PRELIMINARY

The architecture of Large Multimodal Models (LMMs) typically consists of three key components, *i.e.*, a visual encoder, a projector, and a Large Language Model (LLM). As shown in Fig. 2, given a pair of image and text prompt, a visual encoder like CLIP-ViT (Radford et al., 2021), first extracts visual features from the image into a set of visual tokens. To bridge the feature gap between the visual and language component, these tokens are then transformed by the projector to be embedded into the text feature space, which derives $T_v$ for providing the visual context. Meanwhile, the tokenizer of LLM generates discrete text embeddings $T_t$ for the text prompt via the pre-defined vocabulary. Then, the visual tokens $T_v$ and text tokens $T_t$ are fed into the LLM together to predict the answer in an auto-regressive manner. Visual token pruning removes unimportant visual tokens in the forward process during the prefilling stage to accelerate inference. Then, during the decoding stage, retained tokens are attended to by the output token via the KV cache (Pope et al., 2023).

### 3.2 UNBIASED RELEVANCE ESTIMATION (URE)

**Rethinking the attention-based estimation.** To effectively perform visual token pruning for LMMs, it is crucial to develop a metric that accurately assesses the relevance of visual tokens to the text

prompt, allowing for a clear distinction of their importance. Once the tokens are ranked by importance, the least significant ones can be pruned with minimal information loss. Previous research has utilized attention scores of the final input token as a useful metric, as they reflect the attention directed toward visual tokens during generation (Chen et al., 2024b; Dai et al., 2024; Li et al., 2024c). However, as shown in Fig. 1.(a), we observe a significant attention bias in this attention-based metric. In many cases, visual tokens closer to the bottom-right corner consistently receive higher attention scores than others, regardless of their content.

To investigate this, we firstly examine the attention mechanism through which visual tokens are processed. Specifically, for the $i$-th token in the input sequence, we denote its query feature, key feature, and rotary position embedding (Su et al., 2024) as $\boldsymbol{q}_i$, $\boldsymbol{k}_i$, and $\boldsymbol{R}_i$, respectively. The attention scores that visual tokens received from the last token in the prompt can be derived by $\boldsymbol{a}_i = \frac{\exp(\boldsymbol{q}_n^T \boldsymbol{\mathcal{R}}_{i-n} \boldsymbol{k}_i)}{\sum_j \exp(\boldsymbol{q}_n^T \boldsymbol{\mathcal{R}}_{j-n} \boldsymbol{k}_j)}$, where $n$ denotes the last token position and $\boldsymbol{\mathcal{R}}_{i-n} = \boldsymbol{R}_n^T \boldsymbol{R}_i$. We omit the scaling factor in the calculation for brevity. Then, the averaged score across different heads can be used to estimate the importance of visual tokens.

In the context of visual instructions, two-dimensional visual tokens are flattened to formulate one-dimensional input sequence for LLM, resulting in visual tokens located near the bottom right corner being positioned after in the sequence, as shown in Fig. 2. However, previous work (Su et al., 2024) shows that with the same query and key features, the upper bound of $\boldsymbol{a}_i$ gradually decrease as the relative distance, *i.e.*, $n - i$, increases. This is attributed to the rotary position embedding in attention, which leads to the tendency for long-term decay. Consequently, visual tokens with small relative distances to the target prompt token tend to have higher bounds on their attention scores.

To verify that such bias in position affects the criteria of attention scores, we inspect the variation in attention score of visual tokens when integrating positional embeddings at different positions. Formally, we suppose that the positions of visual tokens range from $p_1$ to $p_2$ and uniformly select eight tokens from this interval for observation. For each token positioned at $p$, we employ each position $\overline{p}$ varying from $p_1$ to $p_2$ to obtain its attention score $\boldsymbol{a}_p$ by

$$\boldsymbol{a}_p = \frac{\exp(\boldsymbol{q}_n^T \boldsymbol{\mathcal{R}}_{\overline{p}-n} \boldsymbol{k}_p)}{\sum_{j \neq p} \exp(\boldsymbol{q}_n^T \boldsymbol{\mathcal{R}}_{j-n} \boldsymbol{k}_j) + \exp(\boldsymbol{q}_n^T \boldsymbol{\mathcal{R}}_{\overline{p}-n} \boldsymbol{k}_p)}. \quad (1)$$

As shown in Fig. 3, it can be observed that with the same visual feature, the attention score shows an increasing trend as the position $\overline{p}$ increases for various visual tokens, especially noticeable in the shallow layers. Such bias prevents the attention scores from reliably indicating the relevance of visual information to the instruction, rendering them an inaccurate criteria for assessing the visual token importance.

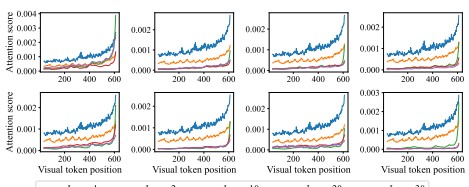

Figure 3: The variation of attention score for different visual tokens under different positions. We uniformly obtain eight visual tokens from the visual position range $[p_1, p_2]$ for observation, *i.e.*, the tokens with the position of $p_1$, $p_1 + \lfloor \frac{p_2 - p_1}{7} \rfloor$, $p_1 + \lfloor \frac{2(p_2 - p_1)}{7} \rfloor$, ..., and $p_2$. They are arranged in order from left to right and from top to bottom. We observe that the attention scores exhibit a notably positive correlation with the position for different visual tokens, which is particularly pronounced in the shallow layers.

**Position-agnostic attention (PA-Att)**. Based on the observations presented above, we propose an unbiased relevance estimation strategy to mitigate the bias inherent in attention-based importance metrics. We introduce a position-agnostic attention mechanism designed to disentangle the interference caused by position embeddings, allowing for more accurate relevance estimation. Specifically, for all visual tokens with the position range of $[p_1, p_2]$, we employ a uniform averaged position $p' = \lfloor \frac{p_1 + p_2}{2} + \frac{1}{2} \rfloor$, ending up with a consistent position embedding $\boldsymbol{R}_{p'}$. When deriving the attention score between the query $\boldsymbol{q}_n$ and visual tokens $\boldsymbol{k}_i$, the relative distance will be $\boldsymbol{\mathcal{R}}_{p'-n}$ for all visual tokens, ensuring being position-agnostic. The final formulation for the position-agnostic attention-based importance estimation for visual tokens can be computed as follows:

$$\boldsymbol{a}_i^* = \frac{\exp(\boldsymbol{q}_n^T \boldsymbol{\mathcal{R}}_{p'-n} \boldsymbol{k}_i)}{\sum\limits_{j \in \boldsymbol{T}_t} \exp(\boldsymbol{q}_n^T \boldsymbol{\mathcal{R}}_{j-n} \boldsymbol{k}_j) + \sum\limits_{j \in \boldsymbol{T}_v} \exp(\boldsymbol{q}_n^T \boldsymbol{\mathcal{R}}_{p'-n} \boldsymbol{k}_j)}. \quad (2)$$

Similarly, we visualize the proposed PA-Att metric in Fig. 1.(a). Our method effectively eliminates position-related attention bias, thus enabling the focus of visual content of tokens and resulting

in more accurate relevance estimation. As demonstrated in our experiments, PA-Att allows the model to retain more informative visual tokens and enhances its ability to perceive visual context for multimodal recognition and reasoning.

### 3.3 Unbiased Token Retention (UTR)

After establishing the importance metric for visual tokens, it is crucial to decide the pruning layers and numbers to remove the less important tokens. We first follow Chen et al. (2024a) and introduce the compression ratio as a measure of the extent of visual token pruning. Specifically, for the $l$-th layer, we denote its number of visual tokens as $N_l$ after pruning. The compression ratio (CR) can be derived by $\mathrm{CR} = \frac{V \cdot L}{\sum_{l=1}^{L} N_l}$, where $V$ means the original visual token count in the input and $L$ is the layer number of LLMs. A larger compression ratio indicates fewer visual tokens after pruning, resulting in a higher inference speed. Given a target compression ratio, existing works typically manually select a single fixed layer for pruning (Chen et al., 2024b;a). However, as shown in Fig. 1.(b), the importance estimated in a single layer cannot adequately reflect the relevance distribution of visual tokens in later layers. Therefore, they suffer from the layer bias, leading to notable visual information loss in the forward process during prefilling. Here, we aim to prune the redundant visual tokens while preserving the overall visual information as much as possible, enabling unbiased token retention.

We start with assessing the degree of visual information loss at each layer with a measurement of pruning loss tolerance. Specifically, given a target compression ratio $c$, for the $l$-th layer, we leverage the importance metric $\boldsymbol{a}^{l-1}$ calculated in the previous layer by Eq. (2) (omitting superscript * for brevity) to retain Top-$\boldsymbol{r}_l$ important visual tokens $\Phi_l$. In this scenario, the visual information loss in the $l$-th layer stems from the pruned $\boldsymbol{r}_{l-1} - \boldsymbol{r}_l$ tokens, $i.e.$, $\Phi_{l-1} \setminus \Phi_l$. Note that during prefilling, the $\boldsymbol{r}_l$ tokens retained at the $l$-th layer should be included in the $\boldsymbol{r}_{l-1}$ tokens reserved at the previous layer, which imposes the constraint of $\Phi_l \subseteq \Phi_{l-1}$. However, it is intractable to obtain the visual information loss under arbitrary $\boldsymbol{r}_l$ and $\boldsymbol{r}_{l-1}$. Considering that the attention scores reflect the information capability of visual tokens (Liang et al., 2022), we thus introduce the assumption that the visual information loss is proportional to the ratio of the attention received by the pruned tokens $\Phi_{l-1} \setminus \Phi_l$ relative to all visual tokens. Based on this, we define the pruning loss tolerance $\boldsymbol{t}_l$ as an approximation of the visual information loss in the $l$-th layer by $\boldsymbol{t}_l = \boldsymbol{\ell}_l \cdot \left( \frac{\sum \{\boldsymbol{a}_k^l \mid k \in \Phi_{l-1} \setminus \Phi_l\}}{\sum_{j \in T_v} \boldsymbol{a}_j^l} \right)$, where $\boldsymbol{\ell}_l$ is the visual information loss when all visual tokens are pruned only at the $l$-th layer. Thus, the objective to minimize whole visual information loss can be tackled by

$$\operatorname*{argmin}_{\boldsymbol{r}_2, \ldots, \boldsymbol{r}_L} \sum_{l=2}^{L} \boldsymbol{t}_l, \quad \text{s.t.} \sum_{l=2}^{L} \boldsymbol{r}_l = \frac{VL}{c} - V, \tag{3}$$

where $\boldsymbol{r}_1 = V$ due to the unavailable metric from the previous layer in the first layer.

Eq. (3) indicates that we can minimize the overall visual information loss by finding an optimal pruning scheme, $i.e.$, $\{\boldsymbol{r}_1, \boldsymbol{r}_2, \ldots, \boldsymbol{r}_L\}$. To this end, we first need to quantitatively derive the $\boldsymbol{\ell}_l$, which, however, is challenging and lacks the standard metric. Here, we follow previous model compression techniques (Frantar & Alistarh, 2022; Frantar et al., 2022) to employ the output feature deviation to assess $\boldsymbol{\ell}_l$. Specifically, we denote the output feature at the last layer for decoding the response as $\boldsymbol{f}_l$ when all visual tokens are pruned only at the $l$-th layer, $i.e.$, $\forall j \in [1, l], \boldsymbol{r}_j = V; \forall j \in [l, L], \boldsymbol{r}_j = 0$. Meanwhile, we obtain original output feature without visual token pruning as $\boldsymbol{g}$. For the $l$-th layer, we leverage the cosine similarity between $\boldsymbol{f}_l$ and $\boldsymbol{g}$ as the metric for the output feature deviation to evaluate $\boldsymbol{\ell}_l$ by $\boldsymbol{\ell}_l = \arccos\left(\frac{\boldsymbol{f}_l^T \boldsymbol{g}}{\|\boldsymbol{f}_l\| \cdot \|\boldsymbol{g}\|}\right)$, where a larger $\boldsymbol{\ell}_l$ indicates a greater impact on the model's generation ability.

Besides, to formulate the constraint of $\Phi_l \subseteq \Phi_{l-1}$, we introduce the binary variable $\boldsymbol{x}_j^l \in \{0, 1\}$ to indicate whether the $j$-th visual token at the $l$-th layer is retained. Therefore, we have $\boldsymbol{r}_l = \sum_j \boldsymbol{x}_j^l$. The objective can thus be transformed into an integer linear programming problem by

$$\min \sum_{l=2}^{L} \boldsymbol{\ell}_l \cdot \left( \frac{\sum_j (\boldsymbol{x}_j^{l-1} - \boldsymbol{x}_j^l) \boldsymbol{a}_j^l}{\sum_j \boldsymbol{a}_j^l} \right), \text{ s.t.} \sum_{l=2}^{L} \sum_j \boldsymbol{x}_j^l = \frac{VL}{c} - V, \tag{4}$$

$$\forall \, l \geq 2 \text{ and } j, \ \boldsymbol{x}_j^l \leq \boldsymbol{x}_j^{l-1}, \ \forall \, \boldsymbol{a}_{j_1}^{l-1} \leq \boldsymbol{a}_{j_2}^{l-1}, \ \boldsymbol{x}_{j_1}^{l-1} \boldsymbol{x}_{j_2}^{l-1} (\boldsymbol{x}_{j_1}^l - \boldsymbol{x}_{j_2}^l) \leq 0. \tag{5}$$

Table 1: **Comparison with SOTA methods on various benchmarks.** CR denotes the compression ratio. * denotes the results with adaptive token numbers in different tasks for PruMerge and PruMerge+, where the target CR cannot be specified.

| Model | Image | CR | VQAv2 | GQA | SciQA | TextVQA | POPE | MME | MMBench | SEED | MMVet | Avg. |
|---|---|---|---|---|---|---|---|---|---|---|---|---|
| LLaVA-1.5-7B (Liu et al., 2024a) | $336^2$ | 100% | 78.5 | 62.0 | 69.6 | 58.2 | 85.9 | 1513.5 | 64.0 | 66.2 | 31.3 | 65.71 |
| PruMerge (Shang et al., 2024) | $336^2$ | 1600% | 66.2 | 51.0 | 68.7 | 53.4 | 64.4 | 1233.2 | 54.3 | 52.7 | 23.8 | 55.13 |
| PruMerge* (Shang et al., 2024) | $336^2$ | 1511% | 66.7 | 51.5 | 69.0 | 53.9 | 67.2 | 1242.3 | 54.6 | 53.2 | 24.2 | 55.82 |
| **UTP** | $336^2$ | 1600% | **71.7** | **54.4** | **69.5** | **55.0** | **74.2** | **1368.8** | **60.7** | **57.5** | **29.4** | **60.09** |
| PruMerge+ (Shang et al., 2024) | $336^2$ | 400% | 73.6 | 56.9 | 69.1 | 54.7 | 79.3 | 1379.5 | 60.1 | 59.9 | 30.3 | 61.43 |
| PruMerge+* (Shang et al., 2024) | $336^2$ | 374% | 74.6 | 57.4 | **69.2** | 55.0 | 82.4 | 1382.8 | 61.5 | 61.7 | 30.5 | 62.38 |
| FastV (Chen et al., 2024b) | $336^2$ | 400% | 72.4 | 55.2 | 68.9 | 56.6 | 70.4 | 1407.1 | 62.8 | 60.8 | 27.5 | 60.55 |
| LLaVolta (Chen et al., 2024a) | $336^2$ | 400% | 70.3 | 57.4 | 68.8 | 49.8 | 73.9 | 1302.1 | 55.9 | 58.0 | 24.7 | 58.21 |
| VisionZip (Yang et al., 2024) | $336^2$ | 400% | 76.0 | 57.9 | 68.9 | 57.0 | 83.4 | 1447.7 | 62.1 | 62.0 | 33.0 | 63.63 |
| **UTP** | $336^2$ | 400% | **77.6** | **60.8** | 69.0 | **57.7** | **84.6** | **1515.5** | **63.8** | **64.8** | **33.3** | **65.26** |
| LLaVA-1.5-13B | $336^2$ | 1600% | 80.0 | 63.3 | 72.8 | 61.2 | 85.9 | 1523.3 | 68.7 | 68.2 | 35.5 | 67.97 |
| PruMerge (Shang et al., 2024) | $336^2$ | 1600% | 67.5 | 51.9 | 72.7 | 54.9 | 62.6 | 1236.9 | 58.7 | 55.2 | 28.5 | 57.09 |
| PruMerge* (Shang et al., 2024) | $336^2$ | 1511% | 67.9 | 52.0 | 72.7 | 55.1 | 64.3 | 1261.5 | 59.3 | 56.2 | 26.4 | 57.44 |
| **UTP** | $336^2$ | 1600% | **73.9** | **56.2** | **74.8** | **57.0** | **76.1** | **1438.5** | **65.8** | **61.0** | **31.5** | **63.14** |
| PruMerge+ (Shang et al., 2024) | $336^2$ | 400% | 74.7 | 57.2 | 73.4 | 56.8 | 77.8 | 1427.2 | 64.3 | 62.5 | 29.1 | 63.02 |
| PruMerge+* (Shang et al., 2024) | $336^2$ | 374% | 75.8 | 58.1 | 74.0 | 56.5 | 81.2 | 1447.1 | 65.5 | 63.6 | 32.6 | 64.41 |
| FastV (Chen et al., 2024b) | $336^2$ | 400% | 75.9 | 58.9 | **74.2** | 59.1 | 76.5 | 1470.5 | 66.6 | 63.8 | 32.2 | 64.53 |
| LLaVolta (Chen et al., 2024a) | $336^2$ | 400% | 73.4 | 59.7 | 71.6 | 53.1 | 79.8 | 1365.8 | 61.9 | 61.3 | 27.2 | 61.81 |
| VisionZip (Yang et al., 2024) | $336^2$ | 400% | 77.0 | 58.5 | 73.8 | 59.1 | 83.4 | 1437.0 | 67.5 | 64.4 | **36.9** | 65.83 |
| **UTP** | $336^2$ | 400% | **79.0** | **62.1** | 73.2 | **60.2** | **85.6** | **1549.8** | **68.1** | **67.5** | 36.1 | **67.70** |

Eq. (5) constraints that visual tokens retained in the later layers need to be reserved in the previous layers and ensures the tokens left in each layer to be those ranked at the top based on the metric. Solving $x_j^l$ can thus derive the optimal value of $r_l$ for each layer.

We note that in LMMs' inference, $r_l$ for each layer cannot be obtained prior to the visual token pruning in the forward process during prefilling. Therefore, we leverage random samples from the *training set* of LMMs, like LLaVA-Instruct-158K (Liu et al., 2024a), for offline estimation. For a given compression ratio $c$, we adopt the efficient integer programming solver (Huangfu & Hall, 2018) to determine the optimal pruning recipe beforehand, *i.e.* $\{r_1, r_2,...,r_L\}$. The solving process is quick, with only nearly two minutes. The scheme of layer-wise retained token numbers is then applied to the model for inference. In our experiment, we show that such an implementation is effective and robust with small amount of samples, demonstrating the good generalization.

## 4 EXPERIMENTS

### 4.1 MAIN RESULTS

**Implementation details.** We perform evaluation for our method based on LLaVA-1.5 (Liu et al., 2024a), following Chen et al. (2024b); Shang et al. (2024); Chen et al. (2024a). We conduct experiments on diverse academic-task-oriented and visual instruction-following benchmarks for LMMs, including VQAv2(Goyal et al., 2017), GQA(Hudson & Manning, 2019), ScienceQA-IMG (Lu et al., 2022), TextVQA (Singh et al., 2019), POPE (Li et al., 2023), MME (Fu et al., 2023), MMBench (Liu et al., 2023), SEED-Bench (Li et al., 2024a), and MM-Vet (Yu et al., 2023). Due to the spatial redundancy in the output visual tokens of the visual encoder (Shang et al., 2024), we also follow Shang et al. (2024) to leverage the attention scores of CLS token from the penultimate layer of the visual encoder to prune redundant tokens before the LLM. The number of its retained tokens, *i.e.*, $r_1$, is simply set to $\frac{4V}{c}$. Besides, we empirically apply the URE strategy in the first three layers, which well eliminates unimportant visual tokens with notable attention bias and incurs minimal overhead. More details and analyses can be referred to the supplementary. We utilize 100 random samples for estimation in UTR. For the average performance across all tasks, we normalize the score of MME by its maximum value of 2000. To ensure a fair comparison, we evaluate baseline methods that are applied directly to pretrained models without additional training, consistent with our method.

**Results.** As shown in Tab. 1, our method achieves the state-of-the-art performance across different model scales and various benchmarks. For LLaVA-1.5-7B, compared with PruMerge+* and FastV, our UTP achieves average performance improvements of 2.88% and 4.71% under the CR of 400%, respectively. On the VQAv2 and GQA for visual question answering benchmarks, our

Table 2: **Ablation study with URE and UTR strategies.**

| Model | URE | UTR | VQAv2 | GQA | SciQA | TextVQA | POPE | MME | MMBench | SEED | MMVet | Avg. |
|---|---|---|---|---|---|---|---|---|---|---|---|---|
| Baseline | | | 72.4 | 55.2 | 68.9 | 56.6 | 70.4 | 1407.1 | 62.8 | 60.8 | 27.5 | 60.55 |
| UTP | ✓ | | 75.3 | 57.2 | 69.0 | 57.6 | 77.1 | 1474.0 | 62.9 | 62.0 | 29.3 | 62.68 |
| | | ✓ | 76.9 | 59.5 | 68.9 | 57.1 | 83.0 | 1488.9 | 63.8 | 64.2 | 30.9 | 64.31 |
| | ✓ | ✓ | 77.6 | 60.8 | 69.0 | 57.7 | 84.6 | 1515.5 | 63.8 | 64.8 | 33.3 | 65.26 |

Table 3: **Uniform pos.**

| Model | GQA | Text. | MME | Avg. |
|---|---|---|---|---|
| Base. | 55.2 | 56.6 | 1407.1 | 60.7 |
| URE-s | 57.6 | 57.5 | 1465.3 | 62.8 |
| URE-e | 55.8 | 57.3 | 1439.6 | 61.7 |
| URE | 57.2 | 57.6 | 1474.0 | 62.8 |

Table 4: **PA-Att.**

| Model | GQA | Text. | MME | Avg. |
|---|---|---|---|---|
| Base. | 55.2 | 56.6 | 1407.1 | 60.7 |
| No pos | 53.5 | 56.6 | 1314.1 | 58.6 |
| Feat cor | 44.6 | 44.5 | 867.4 | 44.2 |
| URE | 57.2 | 57.6 | 1474.0 | 62.8 |

Table 5: **Sample num.**

| Num | GQA | Text. | MME | Avg. |
|---|---|---|---|---|
| 10 | 60.4 | 57.8 | 1498.9 | 64.4 |
| 50 | 60.6 | 57.8 | 1513.8 | 64.7 |
| 100 | 60.8 | 57.7 | 1515.5 | 64.8 |
| 200 | 60.7 | 57.8 | 1515.4 | 64.8 |

Table 6: **Efficiency.**

| Method | CR | Avg. | Latency |
|---|---|---|---|
| FastV | 200% | 64.3 | $13.2e^{-2}$s |
| PruMerge+ | 200% | 62.3 | $12.9e^{-2}$s |
| LLaVolta | 200% | 62.3 | $13.4e^{-2}$s |
| UTP | 400% | 64.8 | $8.7e^{-2}$s |

UTP significantly outperforms VisionZip and LLaVolta by 1.6% and 3.4%, respectively. Besides, for the benchmarks for instruction-following LMMs, our method effectively preserves the multimodal reasoning capability and surpasses the advanced VisionZip by 1.7% and 2.8% on the MMBench and SEED-Bench, respectively. Moreover, given a more aggressive CR of 1600%, our method also exhibits superior performance, which obtains the average improvements of 4.96% and 4.27% over PruMerge and PruMerge*, respectively. For the larger LMM model of LLaVA-1.5-13B, our method also achieves the best overall performance compared with other methods under different CRs. Notably, under the CR of 400%, our UTP only leads to the performance degradation of 0.27% on average across all tasks, achieving competitive performance compared with the original model.

## 4.2 ABLATION STUDY

We analyze the effectiveness of each strategy in our method. We introduce the FastV as the baseline, which leverages the attention scores towards visual tokens as the importance metric and prunes the tokens at a fixed layer. We conduct experiments based on LLaVA-1.5-7B under the CR of 400%. As shown in Tab. 2, each strategy contributes to a favorable performance enhancement. For what to prune, our URE strategy can accurately assess the importance of visual tokens for instruction-following ability and retain more crucial visual tokens, which leads to an average performance improvement of 2.13% over the baseline. For where to prune, our UTR can minimize the output disturbance caused by token pruning and preserve the valuable visual information across layers more effectively, which achieves the overall performance improvement of 3.76% compared with the baseline. Finally, our UTP method can significantly outperform the baseline, achieving the performance improvement of 4.71% on average across all tasks, well demonstrating the effectiveness and superiority.

## 4.3 MODEL ANALYSES

We perform comprehensive analyses for our method, based on LLaVA-1.5-7B with the CR of 400% across diverse tasks including GQA, TextVQA, and MME, by default.

**Different uniform position embeddings.** In URE, we employ the averaged position $p' = \lfloor \frac{p_1 + p_2}{2} + \frac{1}{2} \rfloor$ for all visual tokens, where $p_1$ and $p_2$ denote their start and end positions, respectively. We investigate the performance variation under different uniform position embeddings. Specifically, we assign $p_1$ and $p_2$ as positions for all visual tokens, which are denoted as "URE-s" and "URE-e", respectively. As shown in Tab. 3, compared with the baseline, our URE with different uniform position embeddings can consistently obtain performance improvements, showing its effectiveness. Besides, we observe that "URE-e" obtains inferior results compared with "URE-s" and "URE". It can be attributed to the greater influence of position embedding on the attention score when using the position $p_2$ closest to the latest token, as shown in Fig. 3. It thus interferes the assessment of visual token importance. In contrast, "URE-s" and "URE" achieve similar superior performance. Thus, we simply adopt the averaged position $p'$.

**Different position-agnostic metrics.** Our PA-att leverages the uniform position for all visual tokens in attention to eliminate the influence of positional information on the importance assessment. To verify its effectiveness, we compare with other strategies that do not incorporate the positional

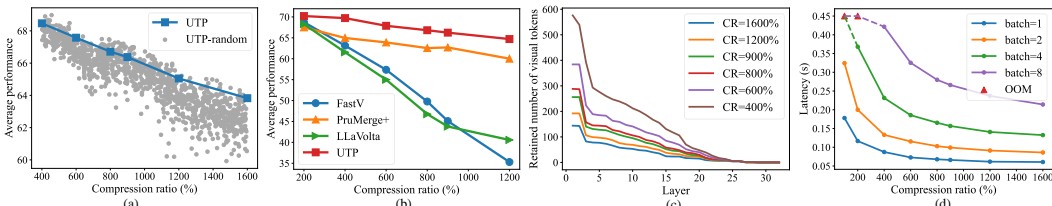

Figure 4: (a) The average performance under various CRs. Our UTP exhibits robust favorable performance compared with "UTP-random" that randomly samples pruning layers and token numbers. (b) Comparison results with others under different CRs. Our method enjoys superiority over others in various scenarios. (c) The retained numbers of visual tokens in each layer in UTR. It shows that LMM favors more visual tokens in shallow layers but notably fewer visual tokens in deep layers. (d) The improved inference efficiency in different scenarios. The latency is measured on a Nvidia RTX3090 GPU. Our method significantly improves the latency under various batch sizes.

information. Specifically, we introduce two baselines: 1) removing the position embedding in the attention, *i.e.*, $a_i = \frac{\exp(q_n^T k_i)}{\sum_{j \in T_t} \exp(q_n^T k_j) + \sum_{j \in T_v} \exp(q_n^T k_j)}$. 2) using the feature correlation as the importance metric, *i.e.*, $a_i = \frac{h_n^T h_i}{||h_n|| \cdot ||h_i||}$, where $h$ denotes the token feature before the attention. They are denoted as "No pos" and "Feat cor", respectively. As shown in Tab. 4, "No pos" and "Feat cor" both obtain inferior results, which indicates the inaccurate importance assessment despite the absence of positions. In contrast, our PA-attn leverages the attention-based importance evaluation. It can well reserve the crucial visual tokens and enable URE to achieve superior performance.

**Effect of sample data.** We leverage random samples to derive the optimal pruning scheme for the model in UTR. We inspect the performance of our method across varying sample sizes. As shown in Tab. 5, we observe that only 10 samples are enough to obtain the satisfactory performance. It indicates that our method works well with minimal sample data, showing its favorable robustness and practicality. It also shows that the LMM favors similar distributions of visual token numbers for different samples. Besides, the performance achieves a saturation point with 100 samples, which serves as an adequate sample size.

**Balance between performance and CR.** To investigate the impact of our method on performance in various scenarios, we inspect the performance variation under different CRs. We also introduce the baseline "UTP-random", which possesses 1000 randomly sampled pruning layers and token numbers. We randomly sample 500 image-text pairs from GQA, TextVQA, and MME for evaluation due to limited resources, respectively. As shown in Fig. 4.(a), our method maintains relatively stable and satisfactory performance under various CRs. Compared with "UTP-random", it exhibits more robust performance and achieves better results in the majority of cases, showing the effectiveness of UTR.

**Comparisons with others under various CRs.** We present the comparison results between ours and others under different CRs in Fig. 4.(b). The average performance is reported based on GQA, TextVQA, POPE, and MME datasets. It can be observed that our method notably outperforms others in various scenarios. These results well show our general efficacy and superiority over others.

**Distribution of retained visual token numbers.** We inspect the retained numbers of visual tokens in UTR under different CRs in Fig. 4.(c). We observe that the shallow layers favor more visual tokens, while the deeper layers retain notably fewer, and may even exclude visual tokens altogether. For the intermediate layers, the number of retained visual tokens exhibits a slow downward trend. It shows the gradual extraction of visual information in the front layers and the high redundancy of visual tokens in the later layers. This inspires that the architecture with fewer visual tokens in deep layers may be more suitable to LMMs.

**Inference efficiency evaluation.** To explore the improvement in inference efficiency of our method, we measure the on-device latencies under different CRs. Following Shang et al. (2024); Chen et al. (2024b), the image-text pair with 576 visual tokens and 64 text tokens is fed into the LMM for analyses. We also follow Shang et al. (2024); Chen et al. (2024b) to employ LLaVA-1.5-7B for inspection and report the prefill time to avoid the influence of output length. As shown in Fig. 4.(d), our method can bring notable inference efficiency gains under different batch sizes and CRs. For example, our method achieves speedups of 2.0× and 2.4× under the batch size of 1 and 2 with

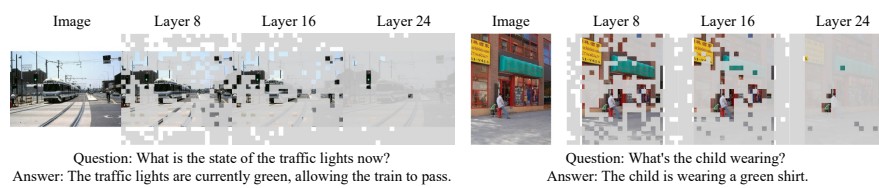

Question: What is the state of the traffic lights now?
Answer: The traffic lights are currently green, allowing the train to pass.

Question: What's the child wearing?
Answer: The child is wearing a green shirt.

Figure 5: Visualization of the token pruning. The images are padded into the square shape in preprocessing for the LMM and the mask indicates the corresponding discarded visual tokens. More examples can be referred to the supplementary.

the CR of 400%, respectively. Along with such impressive efficiency improvements, our method only results in the performance degradation of 0.45% on average across all tasks, as shown in Tab. 1. Besides, our method can reduce the memory footprint during inference and avoid out-of-memory under the batch size of 4 and 8. We also compare with other methods in Tab. 6. It can be observed that our method achieves competitive performance with larger CR, leading to 33% ~ 35% latency reduction over others, showing the potential in practical applications.

**Comparison on other LMMs.** We present the comparison results on Qwen-VL (Bai et al., 2023b), CogVLM (Wang et al., 2025), InternVL2-8B (Chen et al., 2024c), and LLaVA-NeXT-7B (Liu et al., 2024b). We leave out others due to their incomplete compatibility with these LMMs. For Qwen-VL, due to its fewer visual tokens, we adopt the CR of 200%, and for other models, we utilize the CR of 400%. As shown in Tab. 7, our UTP exhibits superior performance across different LMMs. For example, UTP achieves notable improvements of 2.72% and 2.07% on average over FastV for InternVL2-8B and LLaVA-NeXT, respectively, showing the efficacy.

Table 7: **Results on other LMMs.**

| Model | CR | GQA | TextVQA | POPE | MME | Avg. |
|---|---|---|---|---|---|---|
| Qwen-VL | 100% | 54.4 | 60.6 | 85.9 | 1439.6 | 68.23 |
| FastV | 200% | 50.3 | 50.9 | 81.2 | 1393.2 | 63.01 |
| **UTP** | 200% | 53.0 | 57.1 | 85.0 | 1418.2 | **66.50** |
| CogVLM | 100% | 58.6 | 79.2 | 88.1 | 1373.0 | 73.64 |
| FastV | 400% | 47.0 | 51.7 | 73.5 | 1221.0 | 58.31 |
| **UTP** | 400% | 56.4 | 73.9 | 86.9 | 1373.4 | **71.46** |
| InternVL2 | 100% | 60.3 | 77.8 | 84.4 | 1643.7 | 76.17 |
| FastV | 400% | 56.3 | 69.6 | 79.8 | 1561.2 | 70.94 |
| **UTP** | 400% | 58.9 | 72.3 | 82.2 | 1625.5 | **73.66** |
| LLaVA-NeXT | 100% | 64.2 | 61.3 | 86.4 | 1520.3 | 71.98 |
| FastV | 400% | 61.5 | 59.7 | 82.3 | 1447.2 | 68.97 |
| **UTP** | 400% | 63.5 | 59.8 | 86.5 | 1486.8 | **71.04** |

Table 8: **Results on video benchmarks.**

| Model | MSVD | | MSRVTT | | ActivityNet | |
|---|---|---|---|---|---|---|
| | Acc | Score | Acc | Score | Acc | Score |
| Baseline | 68.8 | 3.9 | 57.5 | 3.5 | 43.3 | 3.3 |
| PruMerge | 67.6 | 3.9 | 55.2 | 3.4 | 42.2 | 3.3 |
| FastV | 65.5 | 3.9 | 54.5 | 3.5 | 42.2 | 3.3 |
| LLaVolta | 60.8 | 3.7 | 51.8 | 3.4 | 41.1 | 3.3 |
| **UTP** | 68.5 | 3.9 | 56.4 | 3.5 | 43.1 | 3.3 |

**Generalization on video modality.** To explore the generalization ability of our method to the video modality, we perform evaluation on the video understanding tasks. Experiments are conducted on MSVD-QA (Chen & Dolan, 2011), MSRVTT-QA (Xu et al., 2016), and ActivityNet-QA (Yu et al., 2019), based on the Video-LLaVA-7B (Lin et al., 2023) under the CR of 800%, following Chen et al. (2024b); Shang et al. (2024). As shown in Tab. 8, our method exhibits superior performance over others. Compared with the original model, it can achieve a significant reduction of the number of visual tokens by 8×, with the minimal degradation of 0.3% and 0.2% accuracies on MSVD-QA and ActivityNet-QA, respectively. This shows its favorable generalizability.

**Visualization of token pruning.** To qualitatively show the effectiveness of our method, we visualize the pruned visual tokens in Fig. 5. We show the input image-text pair and the progressive sparsification results. We find that our method can gradually remove the unimportant visual tokens and effectively preserve the crucial visual information for the multimodal perception and reasoning, such as the traffic lights. It also shows the favorable interpretability of our method, *i.e.*, locating the important visual regions which contribute most to the visual instruction following.

## 5 CONCLUSION

In this paper, we propose an Unbiased Token Pruning (UTP) method, to address the challenges of what to prune and where to prune in visual token pruning for LMMs. We introduce the Unbiased Relevance Estimation (URE) strategy to accurately estimate the visual token importance and present the Unbiased Token Retention (UTR) strategy to select the optimal pruning scheme. Thanks to them, our method exhibits competitive performance with favorable efficiency enhancement. Extensive experiments show its superiority over the state-of-the-arts across various tasks.

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

# A  APPENDIX

## A.1  IMPLEMENTATION DETAILS

We follow the evaluation settings of LLaVA (Liu et al., 2024c;a) and conduct all experiments on the Nvidia 3090 GPU with the pytorch framework (Paszke et al., 2019) and transformers library (Wolf et al., 2020). Besides, we utilize the widely adopted KV cache mechanism (Pope et al., 2023) for inference of LMMs. During the prefilling process, only the key and value features of retained tokens are saved in the KV cache of each layer in LLM. During decoding, the output tokens interact directly with the KV cache to calculate the causal attention for generation, avoiding redundant computations of previously processed tokens.

Following Shang et al. (2024); Chen et al. (2024b;a), we evaluate the performance of our method on diverse academic-task-oriented and instruction-following benchmarks for LMMs. Regarding the academic-task-oriented benchmarks, VQAv2 (Goyal et al., 2017) contains images collected from COCO (Lin et al., 2014) and assesses the model's capabilities in visual recognition, spatial reasoning, and language understanding, *etc*. GQA (Hudson & Manning, 2019) consists of samples based on the scene graph structure in Visual Genome (Krishna et al., 2017) and evaluates the performance of models on visual and compositional reasoning. ScienceQA (Lu et al., 2022) includes scientific questions and answers sourced from the textbooks and lectures, evaluating the zero-shot generalization in reasoning pertaining to scientific knowledge. TextVQA (Singh et al., 2019) comprises images embedded with rich textual information, requiring the model to recognize and reason about the textual content. Besides, for visual instruction-following benchmarks, POPE (Li et al., 2023) incorporates both positive and negative objects in the questions with images from COCO (Lin et al., 2014), which requires the model to recognize positive samples and identify negative ones simultaneously. It includes random, common and adversarial splits, with results reported as F1 scores. MME-Perception (Fu et al., 2023) evaluates the overall capabilities of model on several subtasks in both the coarse-grained and fine-grained dimensions. MMBench (Liu et al., 2023) assesses the robustness of model answers with all-round shuffling on multiple choice answers. SEED-Image (Li et al., 2024a) evaluates the generative comprehension ability of LMMs across multiple dimensions with accurate human annotations. MM-Vet (Yu et al., 2023) assesses the performance of model on various multimodal tasks across recognition, reasoning, and math, *etc*.

Besides, we follow Chen et al. (2024b); Shang et al. (2024) to conduct quantitative assessment on the video question answering benchmarks for our method, including MSVD-QA (Chen & Dolan, 2011), MSRVTT-QA (Xu et al., 2016), and ActivityNet-QA (Yu et al., 2019). Specifically, MSVD-QA leverages the MSVD dataset (Chen & Dolan, 2011) which consists of sentence descriptions and video snippets to generate the question-answer pairs with videos. Similarly, MSRVTT-QA utilizes the large-scale open domain video captioning dataset MSRVTT (Xu et al., 2016) to evaluate model's ability to answer questions based on the videos. ActivityNet-QA derives the human-annotated question answering pairs for videos from the ActivityNet dataset (Caba Heilbron et al., 2015), assessing the model performance in the context of long-term spatial-temporal reasoning. Following Maaz et al. (2023); Lin et al. (2023), we leverage the validation splits of MSVD-QA and MSRVTT-QA, and the test split of ActivityNet-QA for evaluation. The pipeline follows previous works (Maaz et al., 2023; Lin et al., 2023) and we leverage the GPT Assistant to assess the accuracy and score of the model's outputs. The score is on a scale of 1-5.

## A.2  MORE MODEL ANALYSES

### A.2.1  COMPARISON WITH PRUNING IN VISUAL ENCODER

In addition to pruning the visual tokens in each layer of LLM during inference, we also follow Shang et al. (2024) to utilize the attention scores of CLS token from the penultimate layer of the visual

Table 9: **Comparison with pruning in visual encoder.** CR denotes the compression ratio.

| Model | Image | CR | VQAv2 | GQA | SciQA | TextVQA | POPE | MME | MMBench | SEED | MMVet | Avg. |
|---|---|---|---|---|---|---|---|---|---|---|---|---|
| LLaVA-1.5-7B | $336^2$ | 100% | 78.5 | 62.0 | 69.6 | 58.2 | 85.9 | 1513.5 | 64.0 | 66.2 | 31.3 | 65.71 |
| CLS | $336^2$ | 1600% | 68.5 | 51.9 | 69.4 | 54.2 | 70.2 | 1283.4 | 58.8 | 54.8 | 27.6 | 57.73 |
| **UTP** | $336^2$ | 1600% | **71.7** | **54.4** | **69.5** | **55.0** | **74.2** | **1368.8** | **60.7** | **57.5** | **29.4** | **60.09** |
| CLS | $336^2$ | 400% | 76.1 | 58.3 | 68.0 | 57.0 | 83.5 | 1435.3 | 63.0 | 62.5 | 32.6 | 63.64 |
| **UTP** | $336^2$ | 400% | **77.6** | **60.8** | **69.0** | **57.7** | **84.6** | **1515.5** | **63.8** | **64.8** | **33.3** | **65.26** |

Table 10: **Comparison results with ToMe.**

| Model | CR | GQA | TextVQA | POPE | MME | Avg. |
|---|---|---|---|---|---|---|
| Baseline | 100% | 62.0 | 58.2 | 85.9 | 1513.5 | 70.44 |
| ToMe | 400% | 58.4 | 53.3 | 83.9 | 1377.7 | 66.11 |
| **UTP** | 400% | **60.8** | **57.7** | **84.6** | **1515.5** | **69.72** |

Table 11: **URE layers.** We inspect accuracy (%) and latency ($10^{-2}$s) under different first $L$ layers.

| Layer ($L$) | 0 | 1 | 2 | 3 | 4 | 8 | 16 | 24 | 32 |
|---|---|---|---|---|---|---|---|---|---|
| Accuracy | 57.1 | 57.2 | 57.4 | 57.7 | 57.7 | 57.8 | 57.9 | 57.9 | 57.9 |
| Latency | 8.66 | 8.69 | 8.71 | 8.74 | 8.77 | 8.88 | 9.09 | 9.31 | 9.53 |

encoder to remove the redundant output tokens of the visual encoder before the LLM in the UTP. Therefore, we introduce this method as a baseline to verify the effectiveness of our UTP. Specifically, given a target compression ratio $c$, we sort the visual tokens according to the CLS token's attention scores of the penultimate layer in the visual encoder and directly retain the top $\frac{V}{c}$ tokens for fed into the LLM. We denote this method as "CLS". We conduct experiments on various benchmarks based on LLaVA-1.5-7B. As shown in Tab. 9, our UTP consistently outperforms the "CLS" method across various tasks. It achieves the significant performance improvements of 2.36% and 1.62% on average under the compression ratios of 1600% and 400%, respectively. These results well highlight the effectiveness and superiority of our method.

### A.2.2 COMPARISON WITH TOME

We also compare our UTP with previous methods for token pruning in Vision Transformers. We present the comparison results with the advanced ToMe (Bolya et al., 2022) based on LLaVA-1.5-7B in Tab. 10. It shows that our UTP also significantly outperforms ToMe across different benchmarks. For example, UTP achieves the notable improvements of 2.4% and 4.4% accuracies on GQA and TextVQA, respectively. The results further verify the efficacy of our method.

### A.2.3 DIFFERENT LAYERS WITH URE

In URE, we employ the uniform position embedding for all visual tokens to mitigate the attention bias. Intuitively, after decoupling the effect of positions on the importance assessment of visual tokens, those tokens that contain less visual information but exhibit notable attention bias can be removed in the initial few layers of LLM. This suggests that in our UTP, adopting URE in the early few layers can yield effective results, meanwhile bringing minimal computational overhead of calculating position-agnostic attention. Therefore, we suppose that the URE is leveraged in the first $L$ layers with the rest layers utilizing the normal attention score, and observe the model's performance and latency results under different values of $L$. $L = 0$ means the visual token pruning without the URE strategy, which is the baseline method. We conduct experiments on TextVQA based on LLaVA-1.5-7B under the compression ratio of 400%. As shown in Tab. 11, adopting the URE only in the first three layers improves the performance by 0.6% compared with the baseline, while only incurring the latency of $8 \times 10^{-4}$s. Considering that further increasing $L$ enhances performance with more additional overhead for position-agnostic attention, we thus empirically incorporate URE in the first three layers by default, for the better balance between performance and efficiency.

Moreover, we inspect the retained number of visual tokens at different positions with the URE strategy adopted in the first three layers. We also compare this with the scenario where the URE strategy is not employed. Specifically, we obtain 100 random samples and visualize the average proportion of retained visual tokens across various positions. As shown in Fig. 6.(a), we can observe that without URE, the proportion of retained visual tokens exhibits a clear positive correlation with their positions due to the attention bias. This results in the retention of tokens that lack important visual information yet are situated at specific positions. In contrast, in the scenario with URE, those tokens located in the bottom right corner with large positions can be effectively removed, indicating the effective

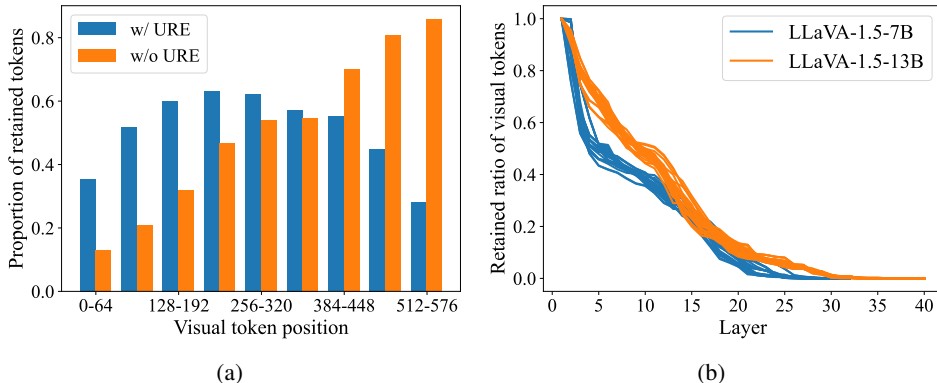

(a)                                                                 (b)

Figure 6: (a) The proportion of retained visual tokens across different positions with and without the URE strategy. It shows that our URE can effectively mitigate the interference of positions and retain more middle regions that usually possess important visual signals. This highlights the efficacy of our method. (b) The distribution of retained token numbers across layers in different samples. It shows that LMMs exhibit similar layer-wise redundancy of visual tokens for different samples.

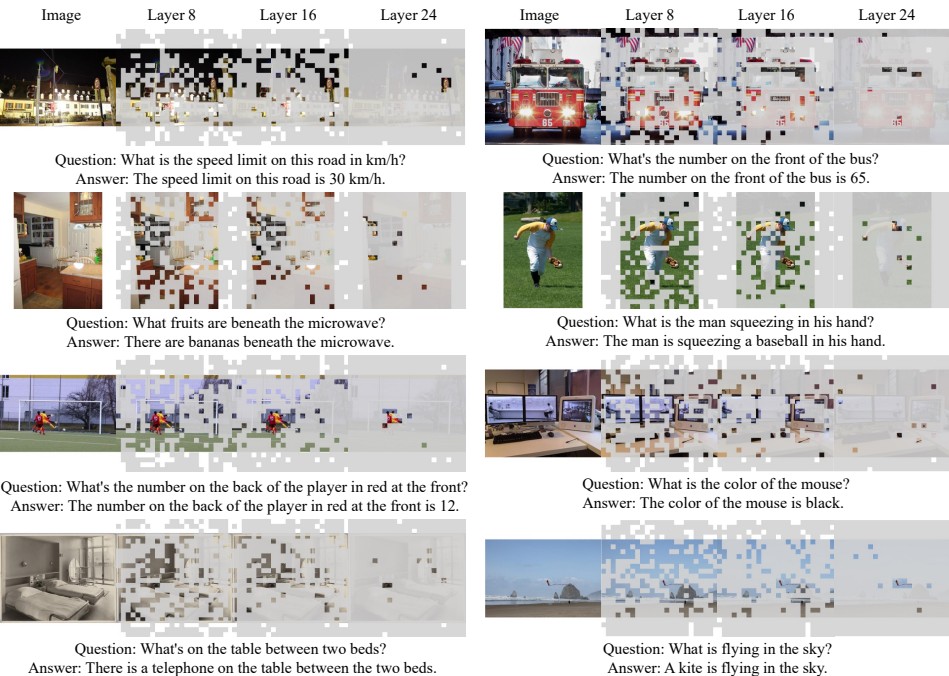

Figure 7: Visualization of our method. The images are padded into the square shape in the pre-processing for the LMM, and the masks indicate the pruned visual tokens. The results show that our method incrementally discards less relevant tokens, allowing for the sharper focus on the most pertinent visual regions.

mitigation of attention bias during the importance assessment. As a result, the tokens in the central region, which typically contain more relevant visual information, can be reserved more frequently, bringing better performance. The comparison further verifies the effectiveness of our strategy.

### A.2.4   RETAINED TOKEN NUMBER DISTRIBUTION IN SAMPLES

We visualize the retained token number distribution across layers for different samples in UTR under the CR of 400% in Fig. 6.(b). It shows that different samples possess similar layer-wise pruning scheme. We attribute such phenomenon to the consistent visual token modeling process in LMM's

layers, *i.e.*, each layer functions similarly and thus exhibits comparable visual token redundancy for different inputs. Furthermore, our UTR can effectively find such scheme with minimal cost, facilitating the mitigation of visual information loss across layers and alleviating the interference of pruning with performance.

### A.3 MORE VISUALIZATION RESULTS

We present more visualization results of token pruning to qualitatively show the effectiveness of our method. As shown in Fig. 7, our method can effectively remove the less relevant visual tokens and retain the crucial visual information for LMM given the instructions. For example, with the first question of "What is the speed limit on this road in km/h?", our method can precisely locate the key position of the speed limit sign and ensure its consistent maintenance. Given the last question of "What is flying in the sky?", our method can effectively identify the important tokens that capture the visual information of the kite and retain them. Besides, in the scenario where the optical character recognition capability is required, with the second question of "What's the number on the front of the bus?", our method can protect the visual tokens corresponding to the digits and help LMM correctly recognize the number. These qualitative results well highlight the efficacy of our method.

### A.4 LIMITATION AND SOCIETAL IMPACT

**Limitation.** Due to limited resources, we do not investigate the performance of our method on larger-scale multimodal models. Besides, exploring the efficacy of our method on models that support other modalities such as audio is also promising and valuable, which we leave as the future work.

**Societal Impact.** The models accelerated by our method can be applied in various scenarios, including image captioning and visual question answering, *etc*. We hope that our method can contribute to these fields by improving efficiency. However, we also recognize the potential misuse of the models, which we will make every effort to prevent.

### A.5 THE USE OF LARGE LANGUAGE MODELS

We only use large language models (LLMs) for polishing writing. All ideation and experiments are conducted by ours.

