# OpenReview forum: "Unbiased Token Pruning for Efficient Large Multimodal Models"
_ICLR.cc/2026/Conference — ICLR 2026 Conference Withdrawn Submission_

### Official Review · Reviewer_Hv6A · 2025-10-25

**Soundness:** 3
**Presentation:** 3
**Contribution:** 3
**Rating:** 6
**Confidence:** 4

**Summary:**

This paper presents a training-free token pruning method (UTP) designed to accelerate Large Multimodal Models (LMMs). The work builds on the well-established approach of using unidirectional attention scores as a proxy for token importance. Within this context, the paper makes a solid contribution by addressing two critical biases that have been under-explored: positional bias in importance estimation (addressed with URE) and layer bias in pruning strategy (addressed with UTR). The extensive experiments demonstrate that UTP achieves state-of-the-art performance and efficiency among existing training-free methods across a wide range of benchmarks. The proposed solutions are effective, and the work is highly practical. However, the paper’s contribution is somewhat limited by its focus on a single paradigm, and the motivations and evaluations could benefit from considering alternative approaches and more comprehensive metrics.

Therefore, my overall stance on this paper is weak accept. If the authors could provide additional details or point out areas I may have overlooked, I would be willing to further increase my score.

**Strengths:**

1. **Effective Solution to Key Problems**: The paper effectively identifies and solves two specific shortcomings—positional and layer bias—within the "attention-as-importance" paradigm. The proposed URE and UTR strategies are novel, direct, and provide effective solutions to these well-defined problems.

2. **Thorough Empirical Validation**: The experiments conducted within the chosen methodology are extensive. The paper convincingly shows that UTP outperforms comparable training-free methods across various models, tasks, and compression ratios. Additionally, the ablation studies effectively break down the contribution of each component.

3. **Practical and Generalizable**: The training-free nature of UTP is a key practical advantage, enabling immediate deployment. The paper convincingly demonstrates that significant efficiency gains can be achieved without the need for costly retraining, which is an appealing feature for practical applications.

**Weaknesses:**

1. **Limited View of "Importance"**: The paper is grounded in the use of unidirectional attention (from text to visual tokens) as a proxy for token importance. While URE mitigates positional bias, the paper does not explore the bidirectional nature of multimodal reasoning. There is no discussion on how visual tokens might influence text understanding, nor is there engagement with works that investigate cross-modal interaction mechanisms to define importance.

2. **Lack of Motivation and Comparison with Trainable Alternatives**: While the focus on the practical benefits of a training-free approach is justified, the rationale for choosing this design over a trainable alternative is not fully explored. The authors could strengthen their argument by addressing the trade-offs between training-free and trainable approaches. For example, could a trainable pruning module learn a more adaptive importance estimation strategy? What is the potential performance ceiling of a trainable counterpart? Including a simple proof-of-concept experiment comparing UTP with a trainable baseline could provide valuable insight. A multi-stage fine-tuning process—first fine-tuning the projector/adapter with the URE/UTR objective, followed by end-to-end fine-tuning of the entire LLM—could help rigorously test this and highlight the performance-computation trade-offs.

3. **Incomplete Computational Cost Analysis**: While the paper focuses on latency (speedup), it does not provide a comprehensive evaluation of efficiency. Other important metrics such as GPU memory footprint reduction, computational savings (FLOPs), and a breakdown of prefill time should be included. Without these metrics, it is difficult to fully assess the efficiency gains and compare UTP with other compression techniques.

4. **Lack of Failure Mode and Limitation Analysis**: While the paper demonstrates the method’s successes, it does not discuss scenarios where UTP might fail or underperform. It would be helpful to explore the limitations of the method. For instance, under what conditions (e.g., very high compression ratios, tasks requiring fine-grained spatial reasoning, or images with scattered critical information) might the method lead to performance degradation? A qualitative analysis of failure cases would help clarify the boundaries of the approach.

5. **Inadequate Coverage of Relevant Work and Contemporary Baselines**: The paper's review of related work and selection of comparison baselines lacks comprehensiveness, particularly for recent advances in training-free token reduction for MLLMs. The literature review and experiments do not engage with several notable contemporary methods published in 2024, such as TokenPacker (Li et al.), Filter, Correlate, Compress (Han et al.), Feather the Throttle (Endo et al.), and Recoverable Compression (Chen et al.).

**Questions:**

Refer to Weakness.

---

### Official Review · Reviewer_Rgvs · 2025-10-26

**Soundness:** 2
**Presentation:** 2
**Contribution:** 2
**Rating:** 4
**Confidence:** 4

**Summary:**

The paper proposes a training-free visual token pruning method for Large Multimodal Models (LMMs), named Unbiased Token Pruning (UTP). The method is composed of two parts: an Unbiased Relevance Estimation (URE) strategy to mitigate positional bias in attention-based importance scoring, and an Unbiased Token Retention (UTR) strategy that uses integer linear programming to determine a static, layer-wise pruning schedule. The authors claim that this approach addresses key deficiencies in prior work related to "what to prune" and "where to prune." Experiments are conducted on several benchmarks, showing favorable performance against a select set of similar pruning methods.

**Strengths:**

- The paper provides a solid diagnosis and an effective solution for the positional bias found in attention-based importance scores within RoPE-based LMMs.
- The formulation of the layer-wise pruning problem as an ILP is a principled approach for deriving a static pruning schedule, which is an improvement over manual or heuristic layer selection in similar methods.
- The ablation study is well-conducted and clearly demonstrates the benefits of the URE and UTR components relative to the chosen baseline.

**Weaknesses:**

- The paper is motivated by the position bias issue observed in previous attention-based token pruning methods; although such bias has been identified in prior works, this paper does not discuss them [1-3].
- It is recommended to include a comparison and discussion with these methods [4-7].
- The paper does not discuss the inherent drawbacks of its static UTR recipe. A single schedule, even if optimized on a calibration set, is unlikely to be universally optimal. The robustness of this recipe to domain shifts (e.g., from natural images to medical or technical diagrams) is not analyzed or discussed, which is a critical practical concern.

If the authors can provide additional evidence demonstrating the superiority and robustness of the proposed method, I would consider increasing the score.







[1] Zhang, Qizhe, et al. "Beyond text-visual attention: Exploiting visual cues for effective token pruning in vlms." Proceedings of the IEEE/CVF International Conference on Computer Vision. 2025. \
[2] Wen, Zichen, et al. "Token Pruning in Multimodal Large Language Models: Are We Solving the Right Problem?." arXiv preprint arXiv:2502.11501 (2025). \
[3] Liu, Xuyang, et al. "Shifting ai efficiency from model-centric to data-centric compression." arXiv preprint arXiv:2505.19147 (2025). \
[4] Arif, Kazi Hasan Ibn, et al. "HiRED: Attention-Guided Token Dropping for Efficient Inference of High-Resolution Vision-Language Models." Proceedings of the AAAI Conference on Artificial Intelligence. Vol. 39. No. 2. 2025. \
[5] Xing, Long, et al. "Pyramiddrop: Accelerating your large vision-language models via pyramid visual redundancy reduction." arXiv preprint arXiv:2410.17247 (2024). \
[6] Wen, Zichen, et al. "Stop looking for important tokens in multimodal language models: Duplication matters more." arXiv preprint arXiv:2502.11494 (2025). \
[7] Ye, Weihao, et al. "Fit and prune: Fast and training-free visual token pruning for multi-modal large language models." Proceedings of the AAAI Conference on Artificial Intelligence. Vol. 39. No. 21. 2025.

**Questions:**

see weakness

---

### Official Review · Reviewer_w9co · 2025-10-26

**Soundness:** 3
**Presentation:** 2
**Contribution:** 2
**Rating:** 4
**Confidence:** 3

**Summary:**

This paper presents Unbiased Token Pruning (UTP), a novel training-free method to accelerate inference in Large Multimodal Models (LMMs). The method tackles two main challenges in visual token pruning: "what to prune" and "where to prune". To address the "what to prune" problem, the authors propose an Unbiased Relevance Estimation (URE) strategy, which introduces a Position-Agnostic Attention (PA-Att) metric to mitigate the positional bias observed in standard attention-based importance scores. For the "where to prune" problem, they introduce Unbiased Token Retention (UTR), which formulates the selection of pruning layers and ratios as an integer linear programming problem. This aims to minimize information loss across layers, moving beyond the common approach of pruning at a single, fixed layer. The authors conduct extensive experiments on the LLaVA-1.5 model and other LMMs across a wide range of benchmarks, demonstrating that UTP outperforms existing state-of-the-art token pruning methods in both performance and efficiency.

**Strengths:**

- The proposed Unbiased Token Retention (UTR) strategy, which uses integer linear programming to determine the pruning schedule across multiple layers, is a more principled and systematic approach compared to the ad-hoc, single-layer pruning used in many existing methods.
- The method is shown to consistently outperform several recent state-of-the-art methods (FastV, PruMerge+, LLaVolta, VisionZip) across a wide array of benchmarks and on different LMM backbones (LLaVA, Qwen-VL, etc.). The performance gains, especially at high compression ratios, are significant.
- The paper includes thorough ablation studies that validate the contribution of each proposed component (URE and UTR). Further analyses on the effect of sample data, the distribution of retained tokens, and efficiency provide a complete picture of the method's behavior.

**Weaknesses:**

- The UTR strategy requires an offline optimization step on a set of calibration samples to determine the pruning recipe. Although the authors state this process is quick (~2 minutes on 100 samples), it introduces a dependency on representative data and an extra step in the deployment pipeline. The sensitivity of the optimal recipe to the choice of this calibration dataset is not deeply explored.
- The pruning loss tolerance t_i depends on l_i, which is the cosine similarity deviation of the output feature. This is a reasonable heuristic, but its connection to the final task performance is indirect. A more direct measure of information loss, if computationally feasible, could potentially yield better results.
- The experiments are primarily focused on 7B and 13B models. While the method is shown to work on several architectures, its performance and the computational cost of the offline ILP step on much larger models (e.g., >70B) are not discussed. The ILP problem size would grow with the number of layers.
- Lack of comparison with some key baselines, such as sparsevlm [1], pdrop [2], dart [3], vispruner [4]

[1] Zhang, Yuan, et al. "Sparsevlm: Visual token sparsification for efficient vision-language model inference." arXiv preprint arXiv:2410.04417 (2024). \
[2] Xing, Long, et al. "Pyramiddrop: Accelerating your large vision-language models via pyramid visual redundancy reduction." arXiv preprint arXiv:2410.17247 (2024). \
[3] Wen, Zichen, et al. "Stop looking for important tokens in multimodal language models: Duplication matters more." arXiv preprint arXiv:2502.11494 (2025). \
[4] Zhang, Qizhe, et al. "Beyond text-visual attention: Exploiting visual cues for effective token pruning in vlms." Proceedings of the IEEE/CVF International Conference on Computer Vision. 2025.

**Questions:**

Please refer to weaknesses.

---

### Official Review · Reviewer_2Dbz · 2025-10-30

**Soundness:** 3
**Presentation:** 3
**Contribution:** 3
**Rating:** 4
**Confidence:** 5

**Summary:**

This paper introduces UTP (Unbiased Token Pruning), a training-free token pruning framework for efficient inference in large multimodal models (LMMs). The authors argue that existing pruning methods (e.g., FastV, PruMerge) suffer from two systematic biases:
1. Attention bias – attention-based importance scores are correlated with token positions (favoring bottom-right visual tokens).
2. Layer bias – fixed-layer pruning fails to capture the varying importance of visual tokens across layers.
To overcome these, UTP proposes two components:
1. Unbiased Relevance Estimation (URE): replaces attention-based importance with a position-agnostic attention metric that removes positional bias.
2. Unbiased Token Retention (UTR): formulates layer-wise pruning as an integer linear programming (ILP) problem that minimizes information loss to determine optimal token counts per layer.
UTP is evaluated on LLaVA-1.5 (7B/13B) and additional models (Qwen-VL, CogVLM, InternVL2, LLaVA-NeXT) across nine benchmarks (VQAv2, GQA, ScienceQA, POPE, MME, etc.).
It achieves 2–5% higher accuracy than state-of-the-art baselines such as FastV and PruMerge, with up to 2.4× speedup and ~33% latency reduction at equivalent FLOPs.

**Strengths:**

1. Identifies and fixes two systemic biases (attention and layer) in token pruning.
2. Tested on six major LMMs including Qwen-VL, CogVLM, and InternVL2, as well as video-LLaVA.
3. Achieves up to 2.4× inference speed-up and ~33 % latency reduction.
4. 2–5 % higher than FastV and PruMerge under identical FLOPs.

**Weaknesses:**

1. Data-dependent offline step: The ILP optimization requires ~100 sample images to estimate layer redundancy; the resulting schedule is tied to the data domain and model.
2. Limited generality: The fixed, pre-optimized ratios hinder adaptivity across unseen domains or evolving architectures — challenging for large-scale serving.
3. No comparison to DivPrune: DivPrune targets the same problem using diversity-based selection, which is published in CVPR2025; UTP should cite or empirically compare to it or state the reason why it is not relevant to compare.
4. Heuristic assumptions: The ILP objective assumes linear correlation between attention weight and information loss.
5. No uncertainty reporting: All results are single-run without variance analysis.

**Questions:**

1. How does the pruning recipe generalize to datasets with different visual statistics?
2. Could the ILP be approximated online to remove data dependence?
3. How sensitive is performance to the number of sampled examples used in ILP estimation?
4. How transferable is the learned pruning schedule between similar architectures?

---

### Note · Authors · 2025-11-14

I have read and agree with the venue's withdrawal policy on behalf of myself and my co-authors.